# Mechanistic target of rapamycin (mTOR) pathway in Sertoli cells regulates age-dependent changes in sperm DNA methylation

**Saira Amir[1†‡], Olatunbosun Arowolo[1†], Ekaterina Mironova[2], Joseph McGaunn[1§], Oladele Oluwayiose[3#], Oleg Sergeyev[2], J Richard Pilsner[3], Alexander Suvorov[1]***

[1]Department of Environmental Health Sciences, University of Massachusetts, Amherst, United States; [2]Belozersky Institute of Physico-Chemical Biology, Lomonosov Moscow State University, Moscow, Russian Federation; [3]Department of Obstetrics and Gynecology, Wayne State University, Detroit, United States

**\*For correspondence:**
asuvorov@umass.edu

[†]These authors contributed equally to this work

**Present address:** [‡]Department of Biosciences, COMSATS University Islamabad, Islamabad, Pakistan; [§]Perelman School of Medicine, University of Pennsylvania, Pennsylvania, United States; [#]NIAID Collaborative Bioinformatics Research (NIH), Bethesda, United States

**Abstract** Over the past several decades, a trend toward delayed childbirth has led to increases in parental age at the time of conception. Sperm epigenome undergoes age-dependent changes increasing risks of adverse conditions in offspring conceived by fathers of advanced age. The mechanism(s) linking paternal age with epigenetic changes in sperm remain unknown. The sperm epigenome is shaped in a compartment protected by the blood-testes barrier (BTB) known to deteriorate with age. Permeability of the BTB is regulated by the balance of two mTOR complexes in Sertoli cells where mTOR complex 1 (mTORC1) promotes the opening of the BTB and mTOR complex 2 (mTORC2) promotes its integrity. We hypothesized that this balance is also responsible for age-dependent changes in the sperm epigenome. To test this hypothesis, we analyzed reproductive outcomes, including sperm DNA methylation in transgenic mice with Sertoli cell-specific suppression of mTORC1 (*Rptor* KO) or mTORC2 (*Rictor* KO). mTORC2 suppression accelerated aging of the sperm DNA methylome and resulted in a reproductive phenotype concordant with older age, including decreased testes weight and sperm counts, and increased percent of morphologically abnormal spermatozoa and mitochondrial DNA copy number. Suppression of mTORC1 resulted in the shift of DNA methylome in sperm opposite to the shift associated with physiological aging – sperm DNA methylome rejuvenation and mild changes in sperm parameters. These results demonstrate for the first time that the balance of mTOR complexes in Sertoli cells regulates the rate of sperm epigenetic aging. Thus, mTOR pathway in Sertoli cells may be used as a novel target of therapeutic interventions to rejuvenate the sperm epigenome in advanced-age fathers.

## eLife assessment

This potentially **important** study addresses the effects of aging on the sperm epigenome and its consequences for reproductive health. The evidence supporting the main claim remains **incomplete**. This study will be of interest to researchers working on aging and reproductive health.

## Introduction

Over the past several decades, a trend toward delayed childbirth has led to increases in parental age at the time of conception. This delayed parenthood is attributed to secular and socioeconomic factors (*Waldenström, 2016*), reproductive technological advancement (*Bray et al., 2006*), and higher levels

of post-graduate education (*Mills et al., 2011*). The suppressive role of overpopulation on reproductive behavior was also hypothesized (*Suvorov, 2021*). Emerging evidence suggests that higher male age contributes to poor pregnancy outcomes (*Hassan and Killick, 2003*), lower odds of live birth (*Horta et al., 2019*), and adverse health of offspring in later life (*Montgomery et al., 2004*; *Puleo et al., 2012*; *Saha et al., 2009*), including increased susceptibility to early development of cancer (*Contreras et al., 2017*), as well as neurodevelopmental and psychiatric disorders such as schizophrenia (*Gratten et al., 2016*) and autism (*Reichenberg et al., 2006*). In experiments with mice it was demonstrated that higher male age is associated with reduced life span in offspring (*Xie et al., 2018*). Increasing evidence links advanced paternal age with altered offspring phenotype via age-dependent changes in the sperm epigenome (*Ashapkin et al., 2023*; *Jenkins et al., 2014*). Published data indicate that age is a powerful factor affecting DNA methylation and other epigenetic markers in mammalian sperm (*Ashapkin et al., 2023*), and age-dependent changes in these epigenetic mechanisms are involved in the regulation of key developmental pathways, including nervous system-related signaling, Wnt, Hippo, mTOR, and Igf1 (*Guo et al., 2021*; *Ma et al., 2020*; *Pilsner et al., 2021*; *Suvorov et al., 2020*; *Wu et al., 2020*; *Xie et al., 2018*; *Yoshizaki et al., 2021*). Despite this evidence, the mechanistic link between age and epigenetic changes in sperm remains unknown.

Final DNA methylation profiles of spermatozoa are a result of epigenetic events that occur during spermatogenesis (*Wu et al., 2015b*), including meiotic divisions of spermatogonia and spermatocytes associated with passive loss of methylation and de novo methylation (*Oakes et al., 2007*). Global methylation drop of around 12–13% occurs in preleptotene spermatocytes and methylation is gradually reestablished during leptotene-pachytene stages so that the final DNA methylation patterns are obtained by the end of pachytene spermatocyte stage (*Gaysinskaya et al., 2018*; *Loukinov et al., 2002*; *Oakes et al., 2007*). A recent rat study identified the transition from elongating spermatids to late spermatids as another stage associated with changes in methylation of more than 5000 DNA regions (*El Omri-Charai et al., 2023*). After fertilization, most parental-specific epigenetic marks of gametes undergo reprogramming to establish totipotency in the developing embryo. However, imprinted loci, certain classes of repetitive sequences, and other genomic loci escape this reprogramming event and contribute to a non-Mendelian form of inheritance as demonstrated

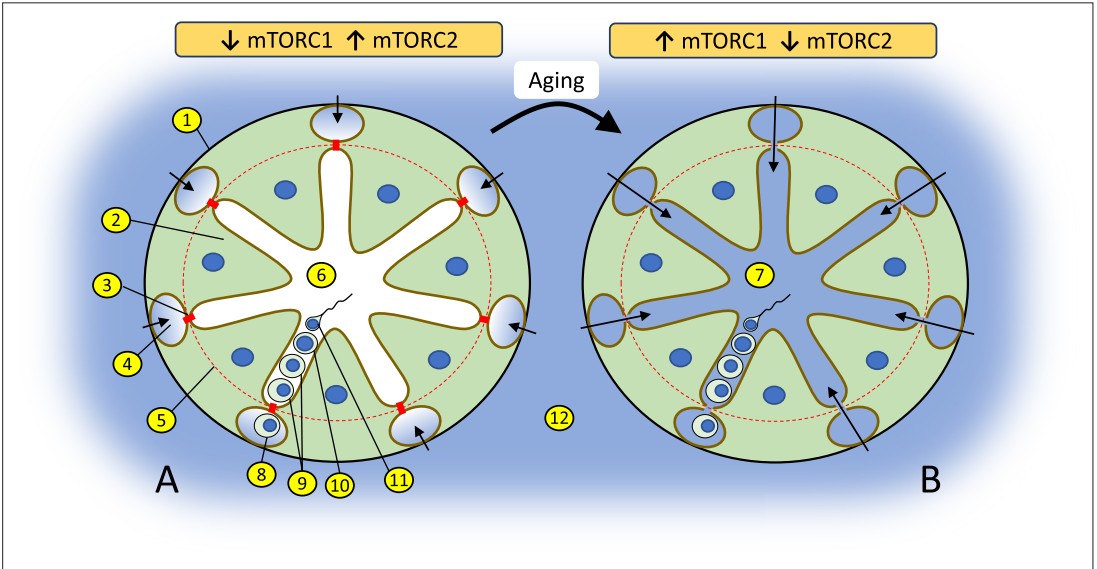

**Figure 1.** Changes in the balance of mTOR complexes and aging affect biochemical conditions of spermatogenesis in the apical compartment of seminiferous epithelium. (**A**) Seminiferous tubule with tight blood-testis barrier (BTB) where interstitial environment (shown in blue) penetrates only to basal compartment of seminiferous epithelium as shown by color and arrows. The tight state of the BTB is promoted by higher activity of mTORC2 over mTORC1 in Sertoli cells. (**B**) Seminiferous tubule with leaky BTB where interstitial environment penetrates to the apical compartment of seminiferous epithelium. BTB disassembly is promoted by higher activity of mTORC1 over mTORC2: (1) basal membrane of a seminiferous tubule; (2) Sertoli cell; (3) contact between Sertoli cells enforced by gap junction, tight junction, and endoplasmic specialization (BTB); (4) basal compartment of the seminiferous epithelium; (5) the dotted red line illustrates structural separation of compartments by the BTB; (6) apical compartment of the seminiferous epithelium; (7) apical compartment in aged organism with biochemically 'noisy' environment due to the leaky BTB; (8) spermatogonia; (9) spermatocytes; (10) spermatid; (11) spermatozoa; (12) interstitial environment.

in humans (*Atsem et al., 2016*; *Denomme et al., 2020*) and rodents (*Oluwayiose et al., 2021*; *Xie et al., 2018*), suggesting that the final sperm epigenome acquired during spermatogenesis is a significant channel for the transfer of inheritable information to the next generation (*Lismer and Kimmins, 2023*).

Epigenetic events during spermatogenesis occur in the unique biochemical environment of the apical compartment of seminiferous epithelium, protected from body fluids by the blood-testis barrier (BTB) – the tightest blood-tissue barrier composed of coexisting tight junctions, basal ectoplasmic specializations, gap junctions, and desmosomes in Sertoli cells (*Figure 1*). The BTB regulates the entry of chemical compounds, including nutrients, hormones, electrolytes, and harmful toxicants, thus creating a unique and highly selective environment for germ cells undergoing epigenetic reprogramming (*Mok et al., 2013a*). In rodents, age-dependent declines in fertility are caused by deterioration of the BTB (*Levy et al., 1999*; *Paul and Robaire, 2013*). Recent studies demonstrate that BTB integrity is determined by the balance between activities of the two complexes of the serine/threonine kinase mechanistic target of rapamycin (mTOR) in Sertoli cells, whereby mTOR complex one (mTORC1) promotes disassembly of the BTB and mTOR complex two (mTORC2) promotes its integrity (*Li and Cheng, 2016*; *Mok et al., 2013a*; *Mok et al., 2013a*; *Figure 1*). mTOR expression is highest in testes out of all tissues in humans (*Fagerberg et al., 2014*) and mice (*Yue et al., 2014*), and the mTOR pathway is recognized today as a major mechanism of longevity and aging regulation (*Papadopoli et al., 2019*; *Weichhart, 2018*). Thus, we hypothesized that the balance of mTOR complexes in Sertoli cells may also play a significant role in age-dependent changes in the sperm epigenome. In the current study we test this hypothesis using transgenic mice with manipulated activity of mTOR complexes in Sertoli cells.

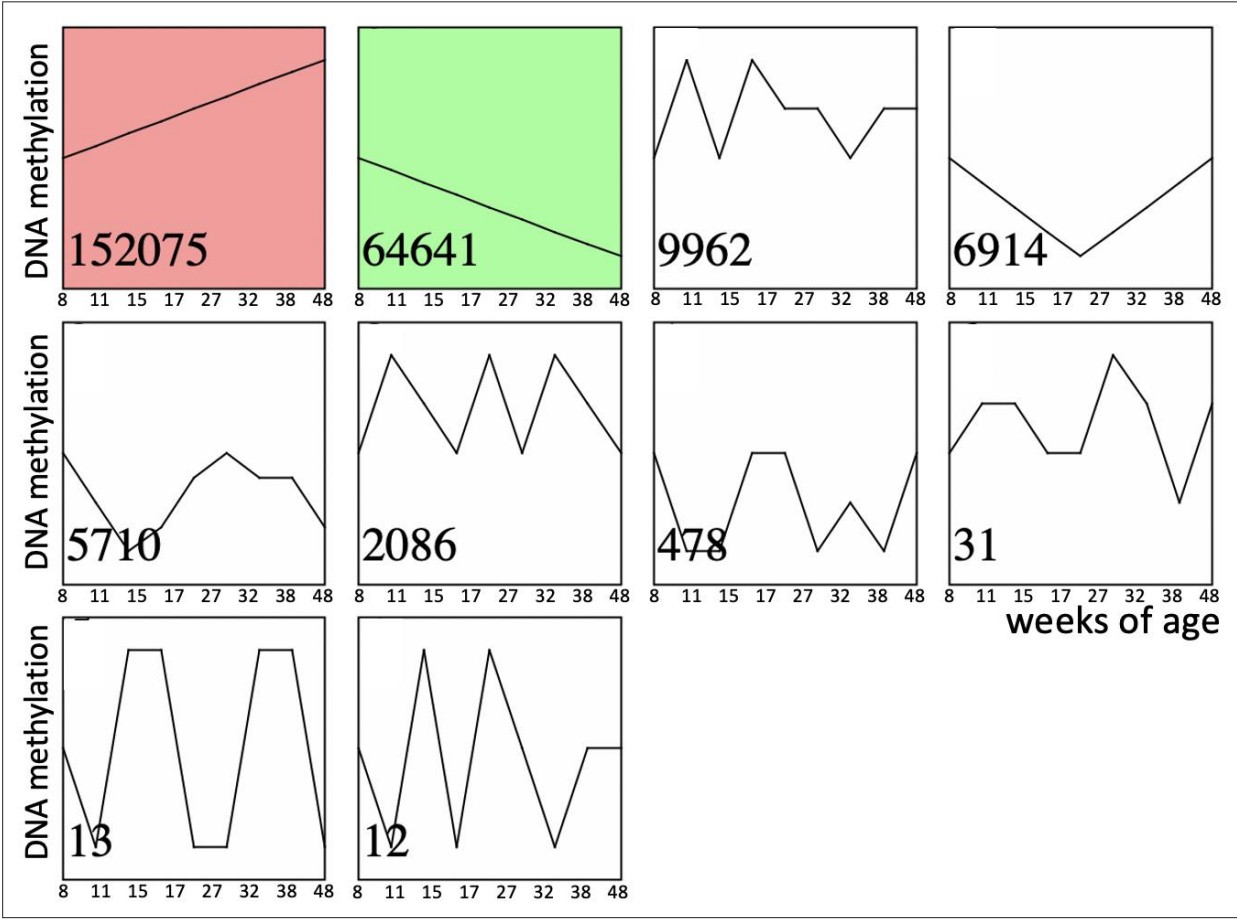

**Figure 2.** Time series of age-dependent DNA methylation change in sperm. Significantly enriched time-series are shown in color. Numbers in the left bottom corner of each time series indicate the number of corresponding CpGs out of total 254,410 CpGs analyzed.

## Results

### Patterns of age-dependent changes in sperm DNA methylation

Recent comprehensive review of age-dependent changes in sperm DNA methylation in humans and animal models concluded that existing evidence is contradictory and it does not provide clear understanding if most of DNA methylation changes in sperm are linearly associated with age or if these relations are more complex (*Ashapkin et al., 2023*). Thus, we characterized age-dependent changes in methylation of 254,410 CpGs sites over eight timepoint covering the period from 56 to 334 postnatal day (PND) using BeadChip methylation arrays and Short-Time Series Expression Miner (STEM) time series analysis which selects temporal profiles independent of the data and identifies significance of enrichment of clusters of similar profiles (*Ernst and Bar-Joseph, 2006*). STEM analysis demonstrated that only two clusters were enriched significantly representing linear increase and linear decrease in methylation over the entire analyzed age-period (*Figure 2*). We repeated this analysis for the top 10% CpGs with highest variance across all ages. The results were the same as analysis done for all CpGs – only clusters of linear age-dependent methylation increase and decrease were enriched significantly (data not shown). These results suggest that within the studied period (56–334 PND), methylation of

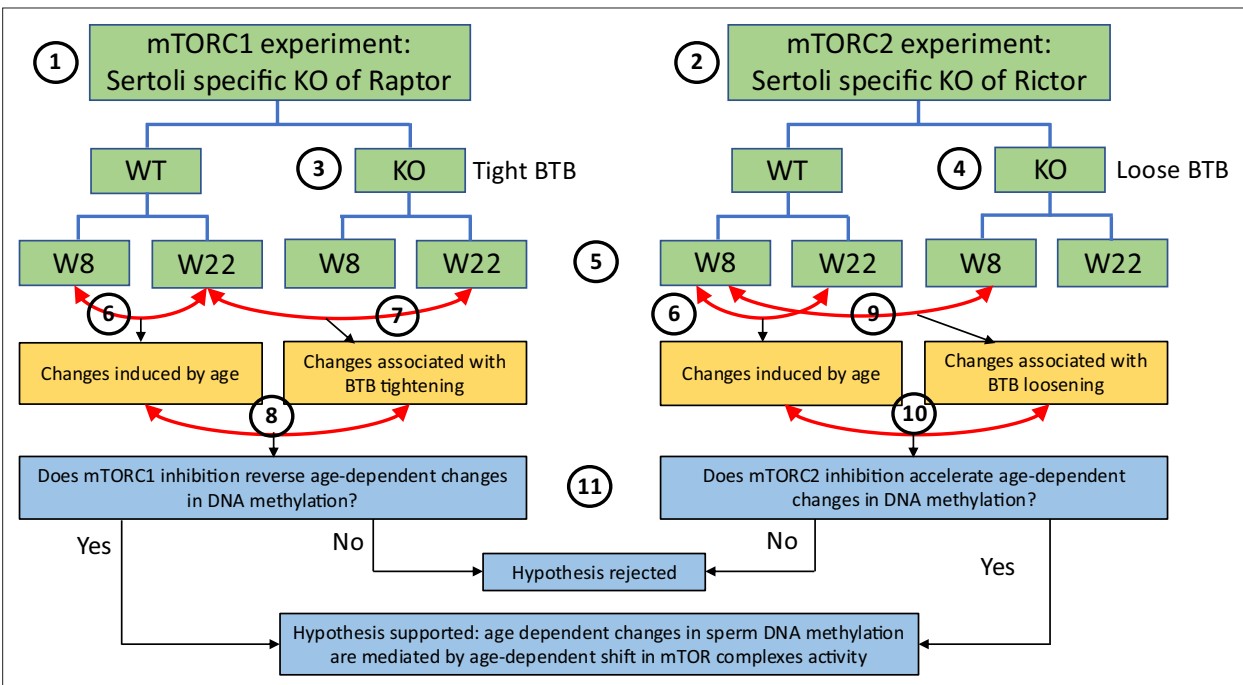

**Figure 3.** Scheme of experimental design for DNA methylation analysis. This study utilized experiments with two transgenic mouse models: (1) one with mTORC1 suppression in Sertoli cells due to the cell-specific knockout (KO) of *Rptor*, and (2) another with mTORC2 suppression in Sertoli cells due to the cell-specific KO of *Rictor*. Suppression of mTORC1 results in tightening of the blood-testis barrier (BTB) (3) and suppression of mTORC2 results in loosening of the BTB (4). DNA methylation changes were analyzed in sperm of each genotype on postnatal weeks 8 and 22 (5). In each experiment, we first identified physiological age-dependent changes in sperm DNA methylation by comparing epigenomes of wildtype (WT) 8-week-old mice and WT 22-week-old mice (6). We further used both experiments to test the hypothesis that age-dependent changes in the sperm epigenome are associated with the age-dependent increase in permeability of the BTB. Specifically, to test this hypothesis using the mTORC1 experiment, we compared physiological age-dependent changes (6) with changes induced by KO (tighter BTB) in older mice (7). Our hypothesis predicts that mTORC1 suppression in older mice will affect age-dependent DMRs in the direction opposite to the one induced by age (8). Similarly, to test our hypothesis using the mTORC2 experiment, we compared physiological age-dependent changes (6) with changes induced by KO (loose BTB) in younger mice (9). Our hypothesis predicts that mTORC2 suppression in young mice due to KO will produce similar effects on age-dependent DMRs as age itself (10). Outcomes of both experiments were used to support or reject our hypothesis (11).

The online version of this article includes the following figure supplement(s) for figure 3:

**Figure supplement 1.** Immunohistochemistry analysis of the efficiency of *Rictor* and *Rptor* knockout (KO) in Sertoli cells.

**Figure supplement 2.** Distribution of CpG per 100 bp regions.

**Figure supplement 3.** Methylation changes per chromosome in mice of different genotype and age.

**Figure supplement 4.** Genomic elements enriched with DMRs associated with age and inactivation of mTOR complexes.

most DNA regions undergoing age-dependent changes in methylation in sperm is associated with age linearly or semi-linearly. Thus, within the studied period, comparison of any two age-groups, distant enough to detect age-dependent change, may be used to characterize sperm methylome aging as well as sperm methylome aging acceleration/deceleration.

## Approach to manipulate the balance of mTOR complexes

To study if shifting the balance of mTOR complexes activity in either direction in Sertoli cells will induce sperm DNA methylation and other reproductive changes concordant with accelerated or decelerated aging, we generated two transgenic mouse models with manipulated mTOR complexes activity. We used the Cre-Lox approach to knockout (KO) critical components of mTORC1 or mTORC2 – *Rptor* and *Rictor*, respectively. In both experiments, *Cre* recombinase was controlled by the anti-Mullerian hormone (*Amh*) promoter to insure KO of Raptor or Rictor in Sertoli cells only (**Figure 3**). We used

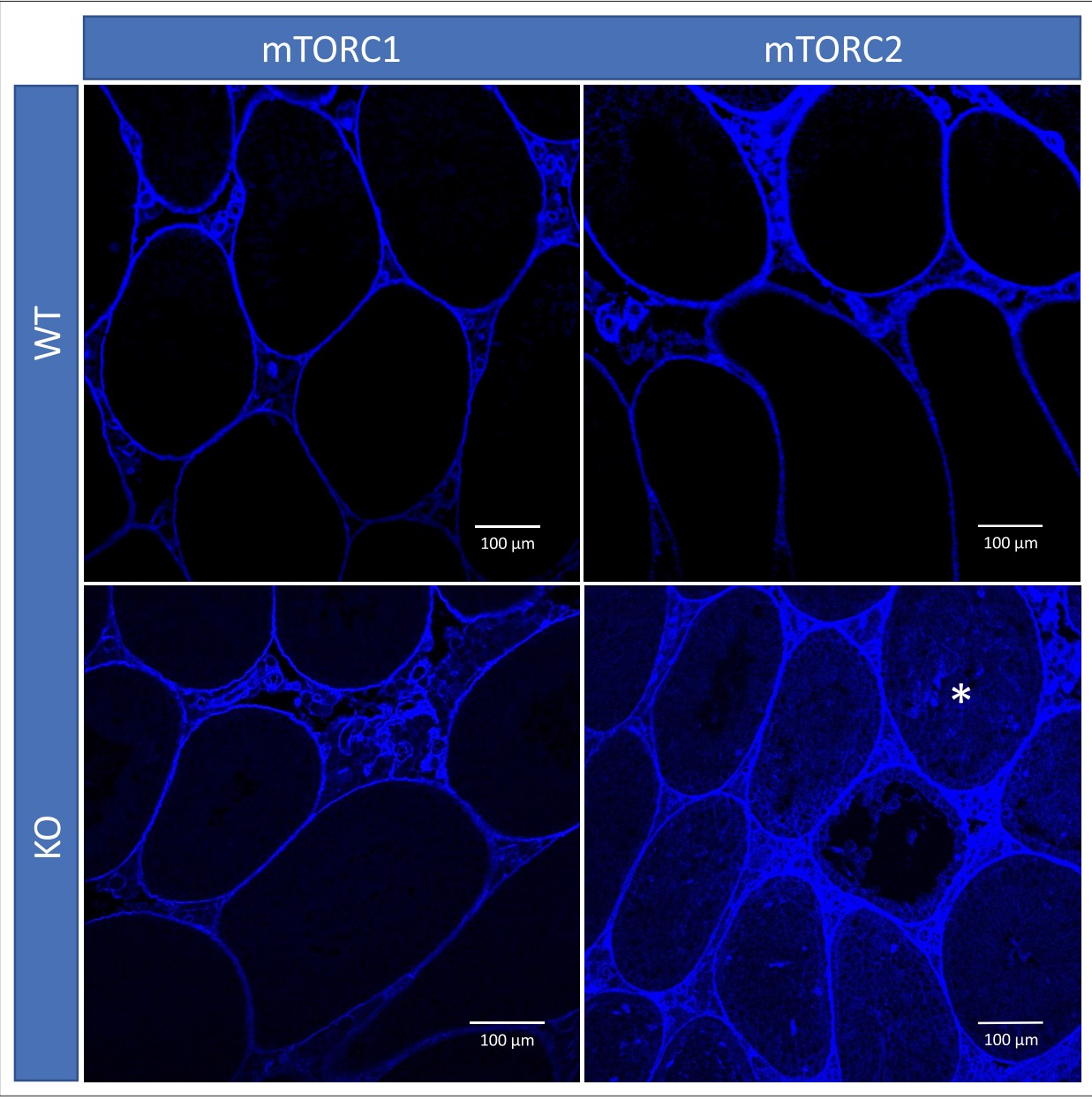

**Figure 4.** Blood-testis barrier (BTB) permeability in 22-week-old mice of different genotypes, representative images. In mice with inactivated mTORC2 (*Rictor* knockout [KO]), biotin tracer (blue) penetrated to the apical compartment of the seminiferous epithelium (asterisk) while in all other genotypes it did not cross the BTB.

immunohistochemistry (IHC) with antibodies for Raptor and Rictor to test efficiency of our genetic manipulations. Indeed, staining of Sertoli cells with Raptor and Rictor antibodies was abolished in corresponding KO animals (*Figure 3—figure supplement 1*). BTB permeability was measured in 22-week-old KO mice and their wildtype (WT) siblings using a biotin tracer assay, where biotin is injected into the testis interstitium in vivo, and its diffusion into the apical compartment of seminiferous epithelium is visualized in histological sections by streptavidin conjugated with a florescent dye (*Meng et al., 2005*). As predicted, in mice with suppressed mTORC2 the BTB was loose enough to allow biotin to migrate to the lumen of seminiferous tubules (*Figure 4*).

## Age-dependent changes in sperm methylome affect methylation of developmental genes

We used reduced representation bisulfite sequencing (RRBS) to analyze DNA methylation in sperm to test if manipulating mTOR complexes balance may accelerate or decelerate epigenetic aging of sperm. For this analysis we first identified regions of DNA that undergo significant changes in methylation in WT animals. To do so, we compared profiles of DNA methylation between 8- and 22-week-old WT mice in each of the two experiments (*Figure 3*). These ages correspond to young pubertal and mature adult stages in humans (*Bell, 2018*; *Flurkey et al., 2007*). The first age (8 weeks) was selected to avoid sperm from the first wave of spermatogenesis be included in the analysis to ensure that observed changes are truly aging dependent rather than associated with different stages of

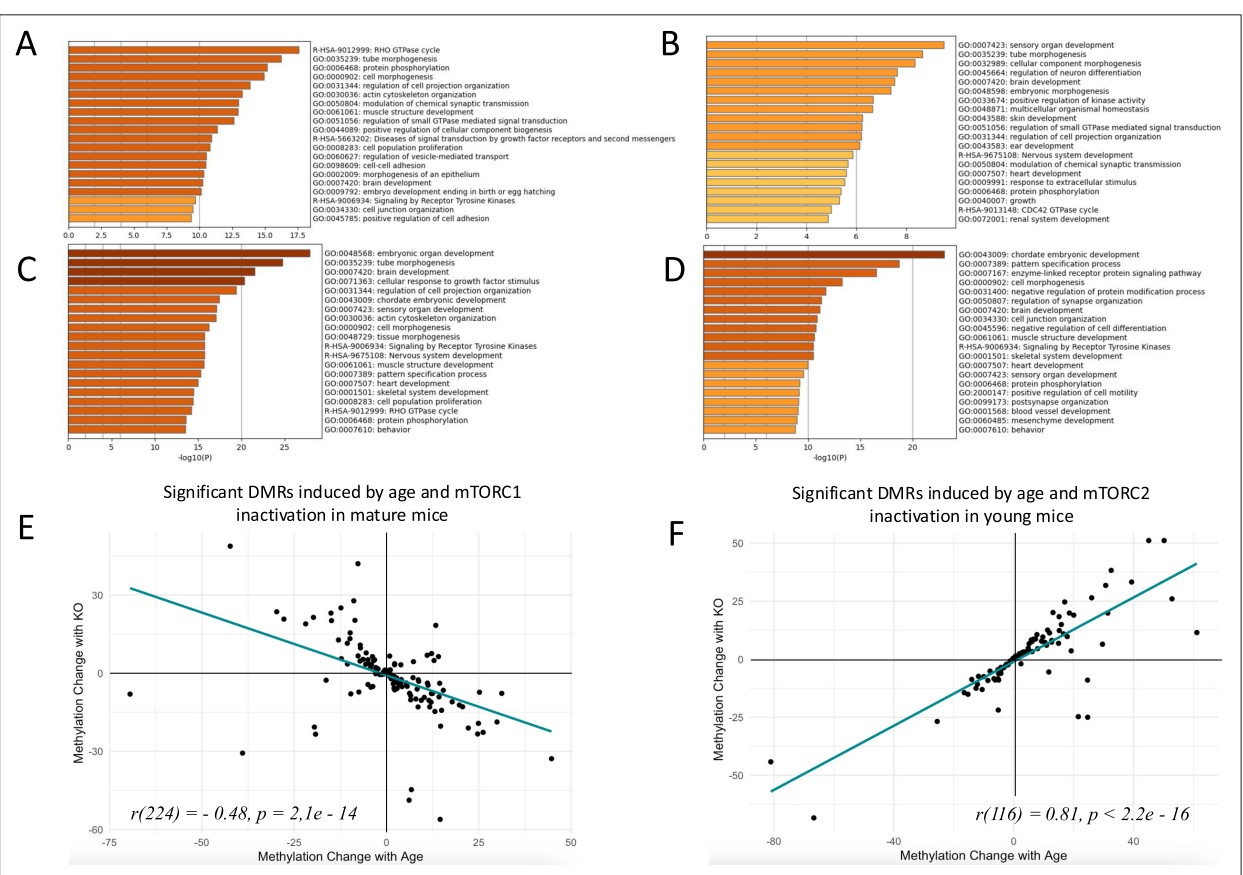

**Figure 5.** Changes in sperm DNA methylation induced by age and manipulation of mTOR pathway in Sertoli cells. (**A, B, C, D**) Top biological categories enriched with genes associated with significant DMRs: (**A**) age-dependent DMRs in wildtype animals in the mTORC1 experiment; (**B**) age-dependent DMRs in wildtype animals in the mTORC2 experiment; (**C**) DMRs induced by mTORC1 inactivation in mature animals; (**D**) DMRs induced by mTORC2 inactivation in young animals. (**E and F**) Changes in methylation of significant DMRs induced by age and mTORC1 inactivation in mature mice (**E**) and by age and mTORC2 inactivation in young mice (**F**).

The online version of this article includes the following figure supplement(s) for figure 5:

**Figure supplement 1.** Changes in methylation of significant DMRs induced by age and genetic manipulation of the mTOR pathway in Sertoli cells.

reproductive system maturation. C57BL/6 mice first have fertile sperm in cauda epididymis at 37 days of age (*Mochida et al., 2019*), and young C57BL/6 mice ejaculate spontaneously around three times per 5 days (*Huber et al., 1980*). Thus, no sperm from the first wave of spermatogenesis may survive in their cauda epididymides to the age of 8 weeks. Methylation changes were analyzed for 100 bp windows containing a minimum of 1 CpG. Average number of CpGs per region was 10.18±0.08 and 10.32±0.09 (mean±SE) in mTORC1 and mTORC2 experiments, respectively (*Figure 3—figure supplement 2*). Significant DMRs were defined as any 100 bp window with methylation difference at FDR<0.05. Using this approach, we identified 1731 age-dependent DMRs in the mTORC1 experiment and 797 DMRs in the mTORC2 experiment, with 79 common DMRs identified in both experiments. In both experiments age-dependent changes were dominated by an age-dependent increase in methylation (*Figure 3—figure supplement 3*). We assigned each DMR to a gene with overlapping gene body or to a closest gene with transcription start site within 5000 bp downstream of the DMR (*Amir et al., 2023*) and used Metascape (*Zhou et al., 2019*) to analyze biological categories associated with differentially methylated regions. Furthermore, we determined genomic elements associated with significant DMRs (*Figure 3—figure supplement 4*). This analysis demonstrated that in both experiments, lists of genes associated with significant age-dependent DMRs are enriched with developmental categories (*Figure 5A and B*), indicative of the possibility that age-dependent changes in sperm DNA methylation may be causative of the transfer of different developmental epigenetic information to the next generation. These findings are concordant with previous research (*Pilsner et al., 2021*; *Suvorov et al., 2020*; *Xie et al., 2018*). Specifically, common highly enriched categories across both experiments included tube morphogenesis, brain development, cell morphogenesis, and other categories relevant to growth, embryo development, development of nervous system, muscles, and others. Enriched molecular mechanisms were dominated by small GTPase-mediated signal transduction and regulation of kinase activity (protein phosphorylation, signaling by receptor tyrosine kinase, and other). The complete results of enrichment analysis can be found at Dryad (*Amir et al., 2023*).

## Decreased activity of mTORC1 'rejuvenates' sperm methylome

At the next step, we analyzed if Sertoli-specific inactivation of mTORC1 affects methylation of age-dependent DMRs. Given that age-dependent changes in the sperm methylome are associated with age-dependent increase in the BTB permeability, we assumed that decreased permeability of the BTB due to genetic manipulation may result in sperm epigenome 'rejuvenation'. Thus, our hypothesis predicts that DNA regions that undergo methylation changes in WT animals with age will undergo the opposite change in older animals with Sertoli-specific inactivation of mTORC1 as compared to WT mice of the same age (*Figure 3*). In other words, DMRs that change with age will be returned to their younger state by mTORC1 inactivation. We identified 2738 significant DMRs induced by mTORC1 inactivation in mature mice (*Amir et al., 2023*) and compared them with age-dependent DMRs in WT animals. Most of the DNA regions induced by genetic manipulation of Raptor undergo hypomethylation – the direction of change opposite to the one induced by age (*Figure 3—figure supplement 3*). The distribution of genic elements associated with DMRs was similar in relation to age and genetic manipulation, suggesting that both factors affect the similar sets of elements (*Figure 3—figure supplement 4*). Next, we analyzed an overlap between significant age-dependent and KO-dependent DMRs in the list of all methylation regions identified in both comparisons. According to this analysis the two lists were highly overlapping with p<0.00001, indicative that age and mTORC1 inactivation in Sertoli cells affect the similar sets of DMRs in sperm (*Figure 5—figure supplement 1*). Finally, we analyzed the direction of change of overlapping DMRs (*Figure 5E*). We observed a strong negative correlation between methylation changes induced by age and changes induced by mTORC1 inactivation (r = –0.48, p=2.1e–14), suggesting that in mature animals, mTORC1 inactivation in Sertoli cells returns age-dependent DMRs to their younger state. The correlation was even higher for DMRs that undergo 10% or higher change in relation to age or genetic manipulation (r=–0.68, p=2.3e–4, *Figure 5—figure supplement 1*).

## Decreased activity of mTORC2 accelerates epigenetic aging of sperm

Similarly, we assumed that if age-dependent changes in the sperm methylome are associated with age-dependent increase in the BTB permeability, then increased permeability of BTB due to Sertoli-specific inactivation of mTORC2 may be associated with accelerated aging of the sperm epigenome.

Thus, our hypotheses predicts that DNA regions that undergo methylation changes in WT animals with age will undergo similar changes in young animals with Sertoli-specific inactivation of mTORC2 as compared with same age young WT mice (*Figure 3*). We identified 1632 significant DMRs induced by decreased mTORC2 activity in Sertoli cells of young mice (*Amir et al., 2023*). The majority of the DNA regions with methylation changes induced by genetic manipulation of Rictor undergo hypermethylation – the direction of change concordant with accelerated sperm aging (*Figure 3—figure supplement 3*). The distribution of genic elements associated with DMRs was similar in relation to age and genetic manipulation, suggesting that both factors affect similar sets of elements (*Figure 3—figure supplement 4*). Further, we conducted an overlap analysis between significant age-dependent and KO-dependent DMRs in the list of all methylation regions identified in both comparisons. According to this analysis the two lists were highly overlapping with p<0.00001, indicative that age and mTORC2 inactivation in Sertoli cells affect the similar sets of DMRs in sperm (*Figure 5—figure supplement 1*). Finally, we analyzed the direction of change of overlapping DMRs (*Figure 5F*). We observed a strong positive correlation between methylation changes induced by age and changes induced by mTORC2 inactivation (r=0.81, p<2.2e–16), suggesting that in young animals, mTORC2 inactivation in Sertoli cells accelerates epigenetic aging of sperm. The correlation was even higher for DMRs that undergo 10% or higher change in relation to age or genetic manipulation (r=0.83, p<2.6e–8, *Figure 5—figure supplement 1*).

## Manipulations of mTOR complexes affect methylation of developmental genes

Given our data demonstrate that DNA methylation changes induced by suppression of mTOR complexes affect age-dependent genes in sperm, it is not surprising that in both experiments (mTORC1 and mTORC2 suppression) the lists of genes associated with significant KO-dependent DMRs were enriched with developmental categories (*Figure 5C and D*), indicative that age-dependent changes in mTOR pathway and in the BTB permeability may be causative of the transfer of altered developmental epigenetic information to the next generation. Specifically, enriched categories common for both experiments included brain development, heart development, chordate embryonic development, pattern specification process, cell morphogenesis, muscle structure development, signaling by receptor tyrosine kinases, skeletal system development, sensory organ development, behavior, and protein phosphorylation. Thus, there is a high concordance of developmental categories enriched with genes associated with sperm methylation changes induced by natural aging (*Figure 5A and B*) and induced by changes in mTOR complexes activity balance (*Figure 5C and D*) (see also Dryad dataset; *Amir et al., 2023*).

## Decreased mTORC2 activity is concordant with aged reproductive phenotype

We next analyzed if Sertoli-specific manipulation of the mTOR pathway may result in reproductive phenotypes concordant with age-related changes. It was demonstrated previously that testes weight and sperm counts decrease with age, while the percent of morphologically abnormal spermatozoa increases with age in humans and laboratory rodents (*Bujan et al., 1988*; *Gunes et al., 2016*). Additionally, mitochondria DNA copy number (mtDNAcn) increases with age in human sperm (*Zhang et al., 2016*) and it was shown recently that sperm mtDNAcn is a novel biomarker of male fecundity (*Rosati et al., 2020*; *Wu et al., 2019*). Thus, to analyze if manipulated mTOR pathway affects known age-dependent reproductive parameters, we analyzed testes size, sperm counts, sperm morphology, and mtDNAcn in mice of all genotypes across four timepoints covering ages that are equivalent to critical reproductive stages in humans: 8 weeks old – young pubertal; 12 and 22 weeks old – mature adult; and 56 weeks old – advanced paternal age (*Bell, 2018*; *Flurkey et al., 2007*). Concordant with our hypothesis, mice with inactivated mTORC2 (permanently 'leaky' BTB) demonstrated significant decreases in testis weight and sperm counts concomitant with increase in morphologically abnormal sperm and mtDNAcn (*Table 1*; *Figure 6A–D*). Inactivation of mTORC1 resulted in testes weight decrease and slight increase in mtDNAcn but did not affect sperm parameters significantly. These results suggest that although permanent changes of the mTOR complexes balance in either direction may have negative consequences for some reproductive parameters, mice with inactivated mTORC2 in Sertoli cells have a reproductive phenotype concordant with accelerated aging.

**Table 1.** Changes in reproductive parameters induced by Sertoli cell-specific knockout (KO) of Raptor (mTORC1 suppression) and Rictor (mTORC2 suppression).

All parameters significantly different (q≤0.05) in KO mice as compared with same age wildtype (WT) counterparts are shown in bold.

| Experiment | Genotype | Age, weeks | Testes weight, mg Mean ± SE | q | Sperm count/field Mean ± SE | q | % Abnormal spermatozoa Mean ± SE | q | Mitochondrial DNA copy number Mean ± SE | q |
|---|---|---|---|---|---|---|---|---|---|---|
| | | 8 | 111±4 | -- | 197±40 | -- | 31±6 | -- | 0.97±0.53 | -- |
| | | 12 | 115±8 | -- | 338±64 | -- | 32±4 | -- | 0.82±0.10 | -- |
| | | 22 | 136±2 | -- | 362±43 | -- | 37±6 | -- | 1.17±0.26 | -- |
| | WT | 56 | 113±4 | -- | 179±22 | -- | 50±7 | -- | 4.38±0.82 | -- |
| | | 8 | 88±6 | 0.062 | 171±24 | 0.725 | 29±5 | 0.867 | 0.93±0.19 | 0.307 |
| | | 12 | 92±5 | 0.064 | 300±38 | 0.743 | 41±2 | 0.126 | **2.67±0.56** | **0.024** |
| | | 22 | **88±7** | **0.006** | 272±10 | 0.169 | 37±4 | 0.950 | 2.60±0.39 | 0.056 |
| mTORC1 suppression | Raptor KO | 56 | **67±5** | **0.003** | 159±26 | 0.728 | 64±6 | 0.568 | 4.28±1.50 | 0.665 |
| | | 8 | 93±3 | -- | 176±6 | -- | 31±7 | -- | 0.72±0.03 | -- |
| | | 12 | 110±7 | -- | 480±60 | -- | 20±2 | -- | 1.10±0.19 | -- |
| | | 22 | 108±7 | -- | 316±41 | -- | 30±6 | -- | 1.30±0.10 | -- |
| | WT | 56 | 107±7 | -- | 285±45 | -- | 42±6 | -- | 5.52±2.08 | -- |
| | | 8 | **70±1** | **0.050** | **91±3** | **0.001** | **71±3** | **0.005** | **1.07±0.12** | **0.017** |
| | | 12 | **78±3** | **0.009** | **193±28** | **0.009** | **67±8** | **0.001** | **1.99±0.43** | **0.016** |
| | | 22 | **84±7** | **0.048** | **184±26** | **0.049** | **63±6** | **0.016** | 2.45±0.34 | 0.143 |
| mTORC2 suppression | Rictor KO | 56 | **66±2** | **<0.001** | **156±34** | **<0.001** | 61±5 | 0.711 | **16.23±5.12** | **0.049** |

## Discussion

An accumulating body of literature indicates that the sperm epigenome responds to a broad range of factors, including paternal diet (*Watkins et al., 2018*), physical activity (*Benito et al., 2018*), stress (*Wu et al., 2016*), metabolic status (*Donkin et al., 2016*; *Wei et al., 2014*), and environmental exposures (*Wu et al., 2017*), among others. Age is the most powerful known factor affecting DNA methylation in mammalian sperm (reviewed at *Ashapkin et al., 2023*), and age-dependent changes in epigenetic mechanisms in sperm are involved in the regulation of major developmental pathways (*Ashapkin et al., 2023*). Importantly, offspring conceived by fathers of advanced age had shorter lifespan (*Xie et al., 2018*). Additionally, the rates of sperm epigenetic aging (DNA methylation and small noncoding RNA) may be modified by exposures to environmental xenobiotics (*Pilsner et al., 2021*; *Suvorov et al., 2020*).

Despite this evidence, the mechanism(s) linking age or other paternal factors with epigenetic changes in sperm were unknown. The results of our experiments suggest that shifts of the balance of mTOR complexes activities in favor of mTORC1 or mTORC2 in Sertoli cells produce changes in sperm DNA methylation matching aging and rejuvenation respectively. Additionally, decreased activity of mTORC2 produced changes in reproductive parameters concordant with aging. Thus, mTOR pathway in Sertoli cells may be used as novel target of therapeutic interventions to rejuvenate the sperm epigenome in advanced-age fathers.

The complete cascade of molecular events that links mTOR complexes activity with rates of sperm epigenome aging remains unclear. The same mechanism, mTOR complexes balance in Sertoli cells, is responsible for BTB permeability regulation (*Li and Cheng, 2016*; *Mok et al., 2013a*; *Mok et al., 2013b*). Given that BTB is responsible for the maintenance of a unique biochemical environment for germ cells undergoing DNA methylation changes (*Gaysinskaya et al., 2018*; *Loukinov et al., 2002*; *Oakes et al., 2007*; *El Omri-Charai et al., 2023*) and its permeability increases with age (*Levy et al., 1999*; *Paul and Robaire, 2013*), these facts taken together provide a strong basis for a hypothesis

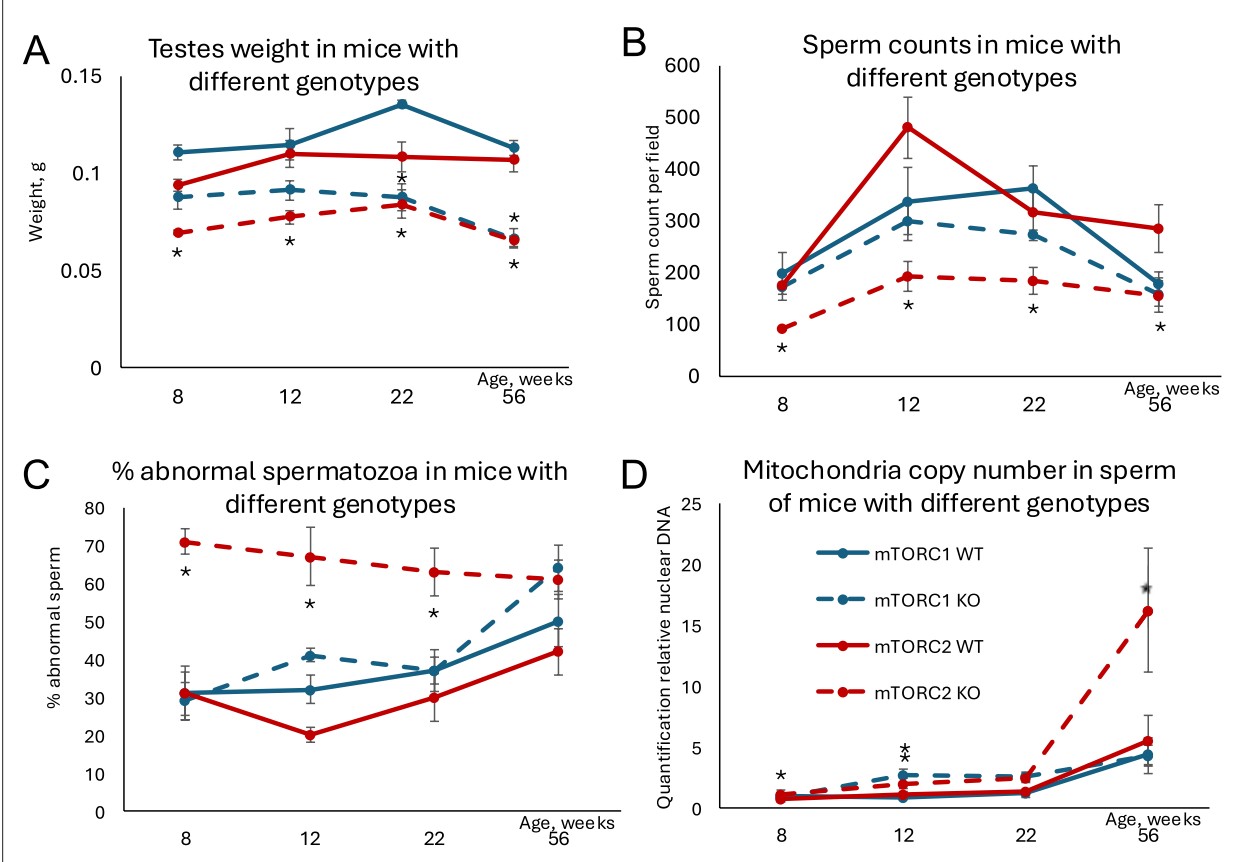

**Figure 6.** Changes in age-dependent reproductive parameters in mice with inactivated mTORC1 or mTORC2 in Sertoli cells. (**A**) Testes weight. (**B**) Sperm counts in sperm smears. (**C**) Percent abnormal spermatozoa. (**D**) Mitochondrial DNA copy number. All data are mean ± SE, n=5–6/age/genotype, *q<0.05 when compared with corresponding wildtype (WT).

that epigenetic aging of sperm results from the loss of BTB integrity. Additionally, research is needed to test this hypothesis.

## Materials and methods
### Identification of age-dependent patterns in sperm DNA methylation

To determine patterns of age-dependent DNA methylation change epididymal sperm was collected from C57BL/6 mice at eight different ages: at PND 56, 80, 107, 118, 190, 225, 265, and 334. Each group consisted of four mice except PND 118 and PND 225 groups which included three mice each. All animals were purchased from Jackson Laboratories. Sperm DNA methylation was determined using Infinium Mouse Methylation BeadChip (MM285K, Illumina) which covers >285,000 methylation sites. The methylation of each CpG in the array was estimated based on the intensity of unmethylated and methylated probes. A total of 288,658 probes were identified and were preprocessed using the sesame function which included background and dye-bias corrections as well as the removal of SNPs and repeated probes in R. The minfi package was used to remove probes that are below background fluorescence levels, adjust the difference in Type I and II probes, and correct for technical variations in the background signals (*Niu et al., 2016*). The ComBat function was used to correct batch effects (*Leek et al., 2012*). This resulted in a total of 254,410 CpGs for downstream analyses. The results are reported in beta values, which is a measure of methylation levels from 0 to 1. The beta values are converted to M-values by log2 transformation, as this has been reported to be a more significant valid approach for differential methylation analysis and better conforms to the linear model homoscedasticity (*Du et al., 2010*). To visualize the sperm DNA methylation pattern with age, a time series analysis of all CpGs was then plotted using the STEM (*Ernst and Bar-Joseph, 2006*). All animal procedures

followed the guidelines of the National Institutes of Health Guide for the Care and Use of Laboratory Animals and the approval for this study was received from the Institutional Animal Care and Use Committee at University of Massachusetts, Amherst (protocol # IACUC: 3615).

## Transgenic mice with Sertoli-specific inactivation of mTORC1 or mTORC2

To inactivate mTORC1, we used a Raptor (component of mTORC1) KO model. To inactivate mTORC2, we used a Rictor (component of mTORC2) KO model. To achieve Sertoli-specific KO, mice with floxed *Rictor* or *Rptor* (Rictor^tm1.1Klg/SjmJ and B6.Cg-Rptor^tm1.1Dmsa/J respectively) were crossed with mice with *Cre* recombinase controlled by the *Amh* Sertoli-specific promoter (129S. FVB-Tg(Amh-cre)8815Reb/J). All mice were purchased from Jackson Laboratories (Stock ##: 020649, 013188, and 007915 respectively). In short, to generate *Rptor* Sertoli-specific KO mice, male B6.Cg-Rptor^tm1.1Dmsa/J mice were paired with female 129S.FVB-Tg(Amh-cre)8815Reb/J mice. In the next round of breeding F1 females were paired with the same B6.Cg-Rptor^tm1.1Dmsa/J males. In F2 offspring around 25% had genotype *Rptor*^flox/flox, and 25% had genotype *Rptor*^flox/flox, *Amh*^cre/+. Animals with these two different genotypes were paired together to produce F3 mice, which were used for sperm collection and reproductive phenotype assessment. F3 male mice were represented by two genotypes: Sertoli-specific KO of *Rptor* (suppressed mTORC1) and WT animals used as controls. The same breeding scheme was used to produce F3 male mice with Sertoli-specific KO of *Rictor* (suppressed mTORC2) and their corresponding WT siblings. All animals used in breeding were 9–10 weeks of age. Breeding was done by pairing one male and two female breeders overnight. Females were inspected for vaginal plugs at 9 am every morning and presence of the plug was considered pregnancy day 1. Pregnant females were housed individually. At weaning on PND 21 all animals were identified via ear-punching, and ear tissue samples obtained from ear-punching were used for genotyping. Genotyping was done using RT-qPCR as recommended by the Jackson Laboratories. All animals were housed in a temperature (23±2°C) and humidity (40±10 %) controlled environment, with a 12 hr light/dark cycle, and food and water available ad libitum. F3 male mice were euthanized at four timepoints: 8, 12, 22, and 56 weeks of age (n≥5 per genotype/timepoint). At each euthanasia testes were collected, weighed, and fixed in modified Davidson's solution and caudal epididymal sperm was collected via swim-up procedure (see below, Sperm collection and DNA extraction). All procedures followed the guidelines of the National Institutes of Health Guide for the Care and Use of Laboratory Animals and the approval for this study was received from the Institutional Animal Care and Use Committee at University of Massachusetts, Amherst (protocol # IACUC: 143 Suvorov_2016–0078).

## Immunohistochemistry

Testes samples collected from 22-week-old mice were fixed in modified Davidson's fluid, dehydrated through a series of alcohols and xylene, and embedded with Paraplast X-tra paraffin (Leica Biosystems) using the Excelsior ES Tissue Processor (Thermo Fisher). Five-micrometer sections were cut on a Microm HM 355S microtome (Fisher) and mounted on Colorfrost Plus slides (Fisher). IHC was performed on a DakoCytomation autostainer, using the envision HRP Detection system (Dako, Carpinteria, CA, USA). Sections were deparaffinized in xylene, rehydrated in graded ethanols, and rinsed in Tris-phosphate-buffered saline (TBS). Heat-induced antigen retrieval was performed in a microwave for 20 min at 98°C in 0.01 M citrate buffer (pH 6) for Rictor antibody and Tris-EDTA (pH 9) for Raptor antibody. After cooling for 20 min, sections were rinsed in TBS and subjected to the primary rabbit polyclonal anti-Rictor (1:100, Abcam, ab70374), rabbit monoclonal anti-Raptor [EP539Y] (1:100, Abcam, ab40768) for 60 and 30 min, respectively. After subsequent washes in TBS, slides were incubated with secondary antibody (DAKO Envision Flex+ anti-rabbit polymer, K800921-2) for 20 min. Immunoreactivity was visualized by incubation with chromogen diaminobenzidine (DAB) (DAKO, K3468) for 10 min. Tissue sections were counterstained with Mayer's hematoxylin (Poly Scientific R&D Corp, S216), dehydrated through graded ethanols and xylene, and coverslipped. Sections were viewed through a Zeiss Axio Observer Z1 inverted light microscope with ZEN imaging software. A minimum of 20 images per slide were captured at 88,000 dpi using the AxioCam 506 color digital camera and analyzed for differences in antibody staining in seminiferous tubules at different stages.

## BTB permeability analysis

The permeability of the BTB was assessed in all WT and KO mice euthanized at 22 weeks of age using a biotin tracer assay as described elsewhere (*Meng et al., 2005*). In short, animals were anesthetized with isoflurane, their testes were exposed, a small opening was created in the tunica albuginea of the right testes, and 50 µl of 10 mg/ml EZ-Link Sulfo-NHS-LC-Biotin (Life Technologies) in PBS containing 1 mM $CaCl_2$ was injected into the interstitium. After 30 min, mice were euthanized, and their testes were immediately removed and fixed in modified Davidson's solution. Testis were embedded in paraffin and 5 µm sections. Deparaffinized slides were incubated with 5 µg/ml Streptavidin Alexa Fluor 405 conjugate in PBS at 25°C for 1 hr, mounted using Diamond Antifade Mountant (P36965), and incubated for 24 hr at room temperature in dark room. The imaging was performed using A1R-SIMe confocal microscope (Nikon) at the light microscopy facility at the Institute of Applied Life Sciences, University of Massachusetts Amherst.

## Sperm collection and DNA extraction

At each euthanasia, the right and left cauda epididymides were incised, cut three times, and incubated at 37°C for 30 min in 1 ml of sperm wash buffer (Cat. # ART1006, Origio, Denmark). After incubation the epididymides were removed, the tube was vortexed for 10 s, and 10 µl of each sample was smeared evenly across the whole surface of the glass slide for microscopic analysis. The remaining samples were used for DNA extraction following the rapid method developed in JR Pilsner's laboratory (*Wu et al., 2015a*). In short, sperm samples were subjected to a one-step density gradient centrifugation over 40% Pureception buffer (CooperSurgical Cat. # ART-2100) at 600×*g* for 30 min in order to remove possible somatic contamination. Sperm cells were homogenized with 0.2 mm steel beads at room temperature in a mixture containing guanidine thiocyanate lysis buffer and 50 mM tris(2-carboxyethyl)phosphine. The lysates were column purified using QiaAMP DNA mini-Kit (QIAGEN, Cat. # 56304) and DNA quality of the eluate was determined using NanoDrop 2000 Spectrophotometer (#E112352; Thermo Scientific, Somerset, NJ, USA). The DNA samples were stored at –80°C.

## Sperm quality analysis

Sperm smears were fixed by immersing in 3% gluteraldehyde in PBS (pH = 7.2) for 30 min, washed briefly in PBS (pH = 7.2) and allowed to dry at room temperature. Smears were stained by 5% aniline blue in 4% acetic acid (pH = 3.5) for 5 min, washed briefly in PBS (pH = 7.2) and allowed to dry at room temperature. Slides were covered with a coverslip using mounting solution incubated at room temperature for 30 min and examined using an inverted light microscope at ×40 magnification. The sperm quantity was assessed by counting spermatozoa in 10 random windows on each slide. For sperm morphology analysis, 200 spermatozoa were counted, and number of abnormal spermatozoa was recorded.

## Mitochondrial DNA copy number

Sperm DNA was used to measure mitochondrial copy number via real-time quantitative PCR using a published protocol (*Machado et al., 2015*). The primers used to calculate relative mitochondrial copy number include a primer pair for the mitochondrial genome (5'- *CTCCGTGCTACCTAAACACC TTATC-3'* and 5'-*GACCTAAGAAGATTGTGAAGTAGATGATG-3')* and another primer pair for nuclear genome for a single copy *Apob* gene (5'-*CACGTGGGCTCCAGCATT-3'* and 5'-*TCACCAGTCATT TCTGCCTTTG-3')*. Triplicate 5 µl real-time PCR reactions, each containing iTaq Universal SYBR Green Supermix (Cat. # 172-5124, Bio-Rad), primers, and cDNA template were loaded onto a 384-well plate and run through 40 cycles on a CFX384 real-time cycler (Bio-Rad Laboratories, Inc). The data were analyzed using the manufacturer's CFX manager software, v3.1. Relative quantification was determined using the ΔΔCq method.

## Library preparation and RRBS

Sperm DNA methylation was analyzed at 8 and 22 weeks of age, with four randomly selected individual samples per age and genotype (*Rictor* KO, R*ictor* WT, *Rptor* KO, *Rptor* WT). Bisulfite conversion was performed on 100 ng of genomic DNA using the EpiTect Fast DNA Bisulfite kit from QIAGEN (Cat.# 59824) and DNA libraries were constructed using the NUGEN Ovation RRBS Methyl-seq System (Cat.# 0353) according to the manufacturer's instructions. Libraries were sequenced on Illumina HiSeq 4000

at the Deep Sequencing Core Facility of the University of Massachusetts Medical School (Shrewsbury, MA, USA) with an average of 22.7 million single-end reads (50 bp) per sample.

## RRBS data analysis

Raw reads were trimmed using TrimGalore (v0.6.6) and a NuGEN-specific adaptor trimming scripts available from GitHub (nugentechnologies/NuMetRRBS). Trimmed reads were mapped using Bismark-Bowtie2 with no mismatch allowed. Methylation counts were called using Bismark extract. Differentially methylated regions were identified using the Methyl kit (v1.24.0) pipeline (*Akalin et al., 2012*). In brief, the genome was tiled into sliding windows and a weighted methylation level was calculated for each window. To minimize error in base calling, we filtered out bases with less than 10× coverage and read counts more than 99.9th percentile and coverage values were normalized using default settings. We used a logistic regression model for p-value calculation subsequently adjusted for multiple comparison (FDR) using the SLIM method for final DMR identification. Individual DMRs were identified for a 100 bp sliding window with a minimum of one CpG in at least three samples out of four in an age/genotype group. DMRs with methylation difference at FDR <0.05 were used to compare different age/genotype groups. To compare the overlap between DMRs induced by age and DMRs induced by KO we used Fisher's exact test. For overlapping DMRs we used Pearson correlation to establish concordance of DNA methylation change induced by aging and genetic manipulations. Genomic ranges were compared using the FindOverlapOfPeaks function available in the chiPpeak-Anno package (*Zhu et al., 2010*). The total number of methylation regions used as a background was determined as a list of 100 bp tiles containing CpGs obtained by intersecting lists from two respective comparisons (e.g. WT8 vs WT22 and WT22 vs KO22 in mTORC1 experiment and WT8 vs WT22 and WT8 vs KO8 in mTORC2 experiment). Spatial genomic annotation was conducted using annotatr package (v1.24.0) (*Cavalcante and Sartor, 2017*) and annotated to the genomic features of the Ensembl genome. Genomic features were compared using the GenomicRanges package (v1.50.2) (*Lawrence et al., 2013*). Each DMR was assigned to the closest gene (<5 kb upstream transcription start site, promoter, 5'UTR, exon, or intron) or intergenic region, and graphs were plotted using ggplot2 package (v3.4.0). We used Metascape (*Zhou et al., 2019*) to analyze biological categories associated with genes annotated to DMRs. For global methylation changes DMRs were analyzed for a 1000 bp sliding window with at least three CpGs and methylation difference >10% at FDR<0.05. The original RRBS data files (fastq) are available via Dryad (*Amir et al., 2023*).

## Statistical analysis

For all sperm parameters differences between KO and corresponding WT age groups were identified using t-test and all p-values were corrected using Benjamini-Hochberg procedure to account for multiple comparison. We considered q (FDR-corrected p)≤0.05 as statistically significant difference. Statistical approaches for bioinformatic analyses are described in the corresponding sections above.

## Acknowledgements

This study was supported by the School of Public Health and Health Sciences, University of Massachusetts Amherst, Dean's Research Enhancement Award to AS.

## Additional information

### Competing interests

Alexander Suvorov: Reports a relationship with ReGENE LLC that includes board membership, equity or stocks, and funding grants. The other authors declare that no competing interests exist.

## Funding

| Funder | Grant reference number | Author |
| --- | --- | --- |
| School of Public Health and Health Sciences, University of Massachusetts Amherst | Dean's Research Enhancement Award | Alexander Suvorov |

The funders had no role in study design, data collection and interpretation, or the decision to submit the work for publication.

## Author contributions

Saira Amir, Olatunbosun Arowolo, Data curation, Formal analysis, Validation, Investigation, Visualization, Methodology; Ekaterina Mironova, Formal analysis, Validation, Investigation, Visualization, Methodology; Joseph McGaunn, Investigation; Oladele Oluwayiose, Formal analysis, Investigation; Oleg Sergeyev, Investigation, Writing – review and editing; J Richard Pilsner, Methodology, Writing – review and editing; Alexander Suvorov, Conceptualization, Resources, Data curation, Formal analysis, Supervision, Funding acquisition, Validation, Investigation, Visualization, Methodology, Writing – original draft, Project administration, Writing – review and editing

## Author ORCIDs

Saira Amir ⓘ https://orcid.org/0000-0002-1864-8572
Alexander Suvorov ⓘ https://orcid.org/0000-0002-2757-5897

## Ethics

All procedures followed the guidelines of the National Institute of Health Guide for the Care and Use of Laboratory Animals and the approval for this study was received from the Institutional Animal Care and Use Committee at the University of Massachusetts, Amherst (protocols # IACUC: 143 Suvorov_2016-0078 and # IACUC: 3615).

Reviewer #1 (Public review): https://doi.org/10.7554/eLife.90992.3.sa1
Reviewer #3 (Public review): https://doi.org/10.7554/eLife.90992.3.sa2
Author response https://doi.org/10.7554/eLife.90992.3.sa3

# Additional files

## Supplementary files
• MDAR checklist

## Data availability

The original sequencing data, processed data and enrichment data were deposited at Dryad: https://doi.org/10.5061/dryad.ncjsxkt0m.

The following dataset was generated:

| Author(s) | Year | Dataset title | Dataset URL | Database and Identifier |
| --- | --- | --- | --- | --- |
| Amir S, Arowolo O, Mironova E, McGaunn J, Oluwayiose O, Sergeyev O, Pilsner JR, Suvorov A | 2023 | Data for: Mouse sperm DNA-methylation changes induced by age and mechanistic target of rapamycin (mTOR) manipulation in Sertoli cells | https://doi.org/10.5061/dryad.ncjsxkt0m | Dryad Digital Repository, 10.5061/dryad.ncjsxkt0m |

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
