## [Editor Report · eLife assessment]

This potentially **important** study addresses the effects of aging on the sperm epigenome and its consequences for reproductive health. The evidence supporting the main claim remains **incomplete**. This study will be of interest to researchers working on aging and reproductive health.

---

## [Referee Report · Reviewer #1 (Public review)]

In the manuscript "Mechanistic target of rapamycin (mTOR) pathway in Sertoli cells regulates age-dependent changes in sperm DNA methylation", the authors proposed to test if the balance of mTOR complexes in Sertoli cells may play a significant role in age-dependent changes in the sperm epigenome. The paper could be of interest and has a good scientific aim but there are too many drawbacks that hamper the initial enthusiasm. All sections need extensive revision. The paper is mostly descriptive without a mechanistic-orientated explanation for the observed results.

Comments on revised version:

I am not sure that the authors have made an attempt to clearly answer the reviewers comments that aimed to improve the quality of the manuscript. It stands as mostly descriptive and with limited interest as it is.

---

## [Referee Report · Reviewer #3 (Public review)]

Summary and Strength:

The manuscript by Amir et al. describes that Sertoli-specific inactivation of the mTORC1 and mTORC2 complex by KO of either Raptor or Rictor, respectively, resulted in progressive changes in blood-testis-barrier (BTB) function, testis weight, and sperm parameters, including counts, morphology, mtDNA content and sperm DNA methylation.

The described studies are based on the hypothesis that a decline of BTB function with increasing chronological age of a male contributes to the DNA methylation changes that are known to occur in sperm DNA of old males when compared to sperm DNA from isogenic young males. In order to demonstrate the relevance of a functioning BTB for the maintenance of sperm methylation patterns, the authors generated mice with genetically disrupted mTORC2 complex or mTORC1 complex in Sertoli cells and determined sperm methylation patterns in comparison to isogenic wild-type males. In line with previously published scientific literature (e.g. Mok et al., 2013; Dong et al, 2015; and others), the manuscript corroborates that a Sertoli-cell specific deletion of mTORC2 caused a loss of BTB function and a progressive spermatogenic defect. The authors further show that sperm DNA is differentially methylated (DMRs) as a consequence of either a mTORC2 disruption (associated with a loss of BTB function) or following a mTORC1 disruption (BTB function either increased or not leaky) when compared to their isogenic age-matched wt controls. Those DMRs overlap partially with changes in sperm DNA methylation that were found when comparing sperm from 8-week males with sperm isolated from 22-week-old male mice.

The authors interpret the observed changes as representative of the sperm DNA methylation changes that occur during normal chronological aging of the male. For an aged control group, the authors use sperm DNA of 22-week-old wild-type mates from the mTORC2 and mTORC2 KO breeding and compare the sperm methylation patterns found in sperm from those 22-week males to 8-week young males, that are intended to represent an old and a young cohort, respectively. DNA methylation analysis indicates that a disruption of mTORC2 (& decrease of BTB function) results in increased DNA methylation of sperm DNA, while a disruption of mTORC1 (and proposed increase of BTB tightness, not shown in the manuscript, though) resulted in increased hypomethylation.

Weaknesses:

While the hypothesis and experimental system are interesting and the data demonstrating the relevance of the mTORC2 complex for BTB function is convincing, several open questions limit the evidence that supports the hypothesis that the sperm DNA methylation changes seen in old males are caused by BTB failure following an imbalance of mTOR signaling complexes. The major critique points are the lack of a chronologically old group and the choice of 8 weeks & 22 weeks age of age:

- Data illustrating the degree of BTB decline and sperm DNA methylation changes from chronologically "old" male mice is missing. 22-week-old mice are not considered old but are of good and mature breeding age, equivalent to humans in their mid-late twenties. In the manuscript, the 22-week-old wildtype mice show no evidence of BTB breakdown (Figure 3), so why are their sperm used to represent "aged" sperm?

- Adding a group of "old" wild-type mice of 12-14 months of age, which is closer to the end of effective reproduction in mice, more equivalent to 45-59 year-old humans could be used to illustrate that (a) aging causes a marked decrease in BTB function at this time in mouse life, and that this BTB breakdown chronologically aligns with the age-associated DNA hypermethylation seen in old sperm. Age-matched "old" mTORC1 KO, with a (supposedly) tighter BTB barrier, could then be expected to have a sperm DMA methylation profile closer to that of younger wild-type animals. Such data are currently missing. While the progressive testicular decline observed in the mTORC1 KO (Fig.5) could make it difficult to obtain the appropriately aged mTORC1 KO tissues, it is completely feasible to obtain data from chronologically old wild-type males. (The progressive testicular decline further raises the question of what additional defects the KO causes, and how such additional defects would influence the sperm DNA methylation profile.) The addition of data from an old group to the currently included groups could strengthen the interpretation that the observations in the BTB-defective mTORC2 KO mice are modelling an age-related testicular decline, provided that the DMRs seen in the chronologically old group significantly overlap with the BTB-defective changes.

- In the current form, the described differences in sperm DNA methylation are based on comparisons between pubertal mice (8 weeks) and mature but not old adult males (22 weeks), while a chronologically "old" group is missing from the data sets and comparisons. Thus, it appears that the described sperm methylation changes reflect developmental changes associated with normal maturation and not necessarily declining sperm quality due to aging. (Sperm obtained from 8-week-old mice likely were generated, at least in part, during the 1st wave of spermatogenesis, which is known to differ from the continuously proceeding spermatogenesis during the remained of the mature life. During the 1st wave of spermatogenesis, Sertoli cells are known to undergo gene expression changes which could contribute to varying degrees of BTB function, and thus have effects on the sperm DNA methylation profiles of such 1st wave sperm.)

- It is unclear why the aging-related DMRs between the 8 and 22-week-old wild-type mice vary so dramatically between the two wild-type groups derived from the mTORC1 and the mTORC2 breeding (Fig. S4). If the main difference was due to mTORC1 or mTORC2 activity, both wildtype groups should behave very similarly. Changes seen in a truly "old" mouse (e.g. 20 weeks to 56 weeks), changes in "young mTORC1" and in "old mTORC2" are missing. How do those numbers and profiles compare to the shown samples?

Comments on latest version:

The rebuttal letter and public response indicate the authors' reluctance to consider the limitations of their study, i.e. having chosen chronologically young animals to demonstrate a sperm aging effect and indicate that they are not willing to include adequate controls.

Since there is no evidence that mice at this young age have a deteriorating blood-testis-barrier (indeed, normal intact BTB is clearly visible in the figures included in this study from animals of the relevant age group), the whole central hypothesis that the study is built upon (i.e. that increasing age causes deteriorating BTB integrity which in turn causes age-related changes in sperm DNA methylation), appears irrelevant or invalid.

The authors' claim that age-related DNA methylation changes in sperm occur in linear fashion and that the changes are somewhat proportional with chronological age is in stark contrast of the claim that a decline of the BTB in old animals is causative for age-related sperm epigenetic changes, putting the relevance of the whole study in question.

---

## [Author Response]

The following is the authors’ response to the current reviews.

**Reviewer #1 (Public Review):**
In the manuscript "Mechanistic target of rapamycin (mTOR) pathway in Sertoli cells regulates age-dependent changes in sperm DNA methylation", the authors proposed to test if the balance of mTOR complexes in Sertoli cells may play a significant role in age-dependent changes in the sperm epigenome. The paper could be of interest and has a good scientific aim but there are too many drawbacks that hamper the initial enthusiasm. All sections need extensive revision. The paper is mostly descriptive without a mechanistic-orientated explanation for the observed results.Comments on revised version:I am not sure that the authors have made an attempt to clearly answer the reviewers comments that aimed to improve the quality of the manuscript. It stands as mostly descriptive and with limited interest as it is.

We are thankful to the reviewer for agreeing to review our revised manuscript. Unfortunately, we completely disagree with the evaluation provided by the reviewer. Research on sperm DNA methylation experienced a significant rise of interest in the current century and by now more than 2000 papers have been published. Although it was demonstrated that the sperm DNA methylome may be affected by almost every factor analyzed, no study was published to identify molecular mechanisms that may link these factors with the sperm epigenome. Our study is the FIRST to identify such a mechanism (mTOR complexes balance in Sertoli cells). More so, we demonstrated experimentally that manipulations of this mechanism allow regulation of the rates of epigenetic aging of sperm in both directions (accelerate aging or rejuvenate). Thus, our study provides a mechanistic background for the development of therapeutic interventions that may target sperm epigenome.

We acknowledge that our study does not provide the full cascade of events linking the balance of mTOR complexes in Sertoli cells with the sperm DNA methylome. It suggests, however, the most plausible event next in a cascade (BTB permeability changes). Our group is working on this question now and we hope to provide the answer soon in a separate study. Even after that, we will be far from understanding the complete chain of molecular events that links mTOR and sperm methylome. It may take many years and significant effort of many research groups to dissect the whole cascade. It is worth mentioning that understanding of a complete cascade involved in pathology is not needed to develop efficient therapies if the critical nodes are known. For many common drugs (e.g. metformin) we do not know the full chain of molecular mechanisms but use them successfully.

Thus, we believe that our study is mechanistic as it identified a critical mechanism manipulation of which allows experimental aging and rejuvenation of the sperm methylome. Additionally, it generates new mechanistic questions and hypotheses to be answered in the future.

**Reviewer #3 (Public Review):**
Summary and Strength:The manuscript by Amir et al. describes that Sertoli-specific inactivation of the mTORC1 and mTORC2 complex by KO of either Raptor or Rictor, respectively, resulted in progressive changes in blood-testis-barrier (BTB) function, testis weight, and sperm parameters, including counts, morphology, mtDNA content and sperm DNA methylation.The described studies are based on the hypothesis that a decline of BTB function with increasing chronological age of a male contributes to the DNA methylation changes that are known to occur in sperm DNA of old males when compared to sperm DNA from isogenic young males. In order to demonstrate the relevance of a functioning BTB for the maintenance of sperm methylation patterns, the authors generated mice with genetically disrupted mTORC2 complex or mTORC1 complex in Sertoli cells and determined sperm methylation patterns in comparison to isogenic wild-type males. In line with previously published scientific literature (e.g. Mok et al., 2013; Dong et al, 2015; and others), the manuscript corroborates that a Sertoli-cell specific deletion of mTORC2 caused a loss of BTB function and a progressive spermatogenic defect. The authors further show that sperm DNA is differentially methylated (DMRs) as a consequence of either a mTORC2 disruption (associated with a loss of BTB function) or following a mTORC1 disruption (BTB function either increased or not leaky) when compared to their isogenic age-matched wt controls. Those DMRs overlap partially with changes in sperm DNA methylation that were found when comparing sperm from 8-week males with sperm isolated from 22-week-old male mice.The authors interpret the observed changes as representative of the sperm DNA methylation changes that occur during normal chronological aging of the male. For an aged control group, the authors use sperm DNA of 22-week-old wild-type mates from the mTORC2 and mTORC2 KO breeding and compare the sperm methylation patterns found in sperm from those 22-week males to 8-week young males, that are intended to represent an old and a young cohort, respectively. DNA methylation analysis indicates that a disruption of mTORC2 (& decrease of BTB function) results in increased DNA methylation of sperm DNA, while a disruption of mTORC1 (and proposed increase of BTB tightness, not shown in the manuscript, though) resulted in increased hypomethylation.Weaknesses:While the hypothesis and experimental system are interesting and the data demonstrating the relevance of the mTORC2 complex for BTB function is convincing, several open questions limit the evidence that supports the hypothesis that the sperm DNA methylation changes seen in old males are caused by BTB failure following an imbalance of mTOR signaling complexes. The major critique points are the lack of a chronologically old group and the choice of 8 weeks & 22 weeks age of age:- Data illustrating the degree of BTB decline and sperm DNA methylation changes from chronologically "old" male mice is missing. 22-week-old mice are not considered old but are of good and mature breeding age, equivalent to humans in their mid-late twenties. In the manuscript, the 22-week-old wildtype mice show no evidence of BTB breakdown (Figure 3), so why are their sperm used to represent "aged" sperm?- Adding a group of "old" wild-type mice of 12-14 months of age, which is closer to the end of effective reproduction in mice, more equivalent to 45-59 year-old humans could be used to illustrate that (a) aging causes a marked decrease in BTB function at this time in mouse life, and that this BTB breakdown chronologically aligns with the age-associated DNA hypermethylation seen in old sperm. Age-matched "old" mTORC1 KO, with a (supposedly) tighter BTB barrier, could then be expected to have a sperm DMA methylation profile closer to that of younger wild-type animals. Such data are currently missing. While the progressive testicular decline observed in the mTORC1 KO (Fig.5) could make it difficult to obtain the appropriately aged mTORC1 KO tissues, it is completely feasible to obtain data from chronologically old wild-type males. (The progressive testicular decline further raises the question of what additional defects the KO causes, and how such additional defects would influence the sperm DNA methylation profile.) The addition of data from an old group to the currently included groups could strengthen the interpretation that the observations in the BTB-defective mTORC2 KO mice are modelling an age-related testicular decline, provided that the DMRs seen in the chronologically old group significantly overlap with the BTB-defective changes.- In the current form, the described differences in sperm DNA methylation are based on comparisons between pubertal mice (8 weeks) and mature but not old adult males (22 weeks), while a chronologically "old" group is missing from the data sets and comparisons. Thus, it appears that the described sperm methylation changes reflect developmental changes associated with normal maturation and not necessarily declining sperm quality due to aging. (Sperm obtained from 8-week-old mice likely were generated, at least in part, during the 1st wave of spermatogenesis, which is known to differ from the continuously proceeding spermatogenesis during the remained of the mature life. During the 1st wave of spermatogenesis, Sertoli cells are known to undergo gene expression changes which could contribute to varying degrees of BTB function, and thus have effects on the sperm DNA methylation profiles of such 1st wave sperm.)- It is unclear why the aging-related DMRs between the 8 and 22-week-old wild-type mice vary so dramatically between the two wild-type groups derived from the mTORC1 and the mTORC2 breeding (Fig. S4). If the main difference was due to mTORC1 or mTORC2 activity, both wildtype groups should behave very similarly. Changes seen in a truly "old" mouse (e.g. 20 weeks to 56 weeks), changes in "young mTORC1" and in "old mTORC2" are missing.How do those numbers and profiles compare to the shown samples?Comments on latest version:The rebuttal letter and public response indicate the authors' reluctance to consider the limitations of their study, i.e. having chosen chronologically young animals to demonstrate a sperm aging effect and indicate that they are not willing to include adequate controls.Since there is no evidence that mice at this young age have a deteriorating blood-testis-barrier (indeed, normal intact BTB is clearly visible in the figures included in this study from animals of the relevant age group), the whole central hypothesis that the study is built upon (i.e. that increasing age causes deteriorating BTB integrity which in turn causes age-related changes in sperm DNA methylation), appears irrelevant or invalid.The authors' claim that age-related DNA methylation changes in sperm occur in linear fashion and that the changes are somewhat proportional with chronological age is in stark contrast of the claim that a decline of the BTB in old animals is causative for age-related sperm epigenetic changes, putting the relevance of the whole study in question.

We are thankful to the reviewer for agreeing to review our revised manuscript. We disagree with the evaluation provided by the reviewer, however.

First, the reviewer misinterpreted the hypothesis of the study, although it is formulated in the last sentence of the Introduction: “ … we hypothesized that the balance of mTOR complexes in Sertoli cells may also play a significant role in age-dependent changes in the sperm epigenome.” Instead, the reviewer assigned a different hypothesis to our study (that BTB integrity changes are responsible for age-dependent changes in sperm DNA methylation) and criticized us for not providing clear testing of this hypothesis.

To clarify, we believe that our study provides high-quality testing of OUR hypothesis as we demonstrated experimentally that manipulations of mTOR complexes balance in Sertoli allow acceleration and deceleration of epigenetic aging of sperm. Additionally, our study generated a hypothesis that BTB permeability may mediate the effects of the mTOR pathway on sperm methylome. This second hypothesis is to be tested in the future research.

We also disagree with the reviewer's interpretation of the aging process as an abrupt transition from a young, healthy, and undamaged state to an old, moribund, and damaged state. The whole body of biogerontological knowledge suggests instead steady accumulation of damage over lasting periods of time. For example, this understanding of steady change at the molecular level allowed the development and successful use of epigenetic clock and other molecular clock models, including several variants of sperm epigenetic clocks. These models clearly demonstrate linear or semi-linear accumulation in DNA-methylation changes in various tissues and biological species across the whole lifespan. It is reasonable to assume that BTB permeability decreases with age steadily as well and that in younger animals this decrease may be not easily detected by existing analytical methods. Experimental data showing the dynamics of the BTB deterioration over age do not exist to our knowledge although it was demonstrated that older animals have loose BTB as compared with young. We agree with the reviewer that future studies testing the role of BTB deterioration for sperm methylome aging will need to provide such evidence. It was not the subject of the current study, however.

The following is the authors’ response to the original reviews.

**Reviewer #1 (Public Review):**
In the manuscript "Mechanistic target of rapamycin (mTOR) pathway in Sertoli cells regulates age-dependent changes in sperm DNA methylation", the authors proposed to test if the balance of mTOR complexes in Sertoli cells may play a significant role in age-dependent changes in the sperm epigenome. The paper could be of interest and has a good scientific aim but there are too many drawbacks that hamper the initial enthusiasm. All sections need extensive revision. The paper is mostly descriptive without a mechanistic-orientated explanation for the observed results.Specific comments:(1) The abstract is poorly written. There is a lot of unnecessary introduction that does not provide a rationale for the work. It is not possible to understand the experimental approach or the major data just by reading the abstract. It does not clearly represent the work.

- We have added details of experimental design and results to the abstract and reduced the introductory part of the abstract.

(2) The introduction is somewhat vague and does not provide a clear rationale for the hypothesis. There should be more focus more on the role of mTOR in Sertoli cells that goes far beyond BTB. That will give more focus on mTOR. Then it is important to focus on BTB and mTOR: what is known? What is the gap and how can it be solved? Several relevant references are missed concerning mTOR and Sertoli cells.

- The goal of this study was not to explore all potential roles of mTOR pathway in Sertoli cells, but to test if shifts in the balance of mTOR complexes regulate (accelerate/decelerate) epigenetic aging of sperm. As such, we disagree with the reviewer and consider that the current Introduction provides a focused rational for the study.

(3) The Material and Methods section needs improvement. There is much important information missing. For instance: how many animals were used per group and how was the breeding done? At what age? Statistical analysis should be explained in detail.

- The number of animals was clearly stated in the original manuscript. We have added details of breeding and statistical analysis.

(4) The results description could be improved. It is vague without highlighting how much difference was detected. The results should be numerically described when possible and the differences should be highlighted. A 10% difference may be significant but not biologically relevant. To correctly evaluate the differences it is important to describe them with some degree of detail.

- For all DNA methylation experiments we provide numerical characteristics of methylation changes, including numbers of DMRs, % change, significance, correlation coefficients. We believe that only age- and genotype-associated changes in reproductive parameters were not characterized in our manuscript in detail. We have added Table 1 to provide these numbers.

(5) There is no discussion of the data. The authors just summarize their findings without a comprehensive analysis of the literature and how the effects can be mediated. mTOR interacts with different pathways (mTORC1 and mTORC2 are even mediators of distinct pathways). This would be very relevant to discuss. In addition, there are many study limitations not discussed. There is no clear mechanistic explanation of the way by which the mTOR pathway in Sertoli cells regulates age-dependent changes in sperm DNA methylation. The paper seems preliminary.

- We have added an additional paragraph to the discussion to highlight a potential molecular mechanism that links mTOR pathway with the sperm epigenome.

(6) Figure 1 is too simple and does not provide any schematic support for the text.

- We disagree with the reviewer and believe that the figure represents a good visualization of our hypothesis useful for the perception of the study.

(7) Figure 2 lacks some detail. For instance, how many animals were used for each step?

- Numbers of animals are provided in the text of the paper.

(8) Taking into consideration the roles of mTOR on sperm, particularly mTORC1, it is not clear whether there were any differences in sperm motility.

- We did not assess sperm motility in this study.

**Reviewer #2 (Public Review):**
In this study, the authors hypothesized that the balance of mTOR complexes in Sertoli cells may also play a significant role in age-dependent changes in the sperm epigenome. To test this hypothesis, the authors use transgenic mice with manipulated activity of mTOR complexes in Sertoli cells. These results suggest that the mTOR pathway in Sertoli cells may be used as a novel target of therapeutic interventions to rejuvenate the sperm epigenome in advanced-age fathers.The authors attempt to demonstrate that the balance of mTOR complexes in Sertoli cells regulates the rate of sperm epigenetic aging. The authors have effectively met their research objectives, and their conclusions are supported by the data presented.

- We are very thankful for the positive evaluation of our study.

**Reviewer #3 (Public Review):**
Summary and Strength:The manuscript by Amir et al. describes that Sertoli-specific inactivation of the mTORC1 and mTORC2 complex by KO of either Raptor or Rictor, respectively, resulted in progressive changes in blood-testis-barrier (BTB) function, testis weight, and sperm parameters, including counts, morphology, mtDNA content and sperm DNA methylation.The described studies are based on the hypothesis that a decline of BTB function with increasing chronological age of a male contributes to the DNA methylation changes that are known to occur in sperm DNA of old males when compared to sperm DNA from isogenic young males. In order to demonstrate the relevance of a functioning BTB for the maintenance of sperm methylation patterns, the authors generated mice with genetically disrupted mTORC2 complex or mTORC1 complex in Sertoli cells and determined sperm methylation patterns in comparison to isogenic wild-type males. In line with previously published scientific literature (e.g. Mok et al., 2013; Dong et al, 2015; and others), the manuscript corroborates that a Sertoli-cell specific deletion of mTORC2 caused a loss of BTB function and a progressive spermatogenic defect. The authors further show that sperm DNA is differentially methylated (DMRs) as a consequence of either a mTORC2 disruption (associated with a loss of BTB function) or following a mTORC1 disruption (BTB function either increased or not leaky) when compared to their isogenic age-matched wt controls. Those DMRs overlap partially with changes in sperm DNA methylation that were found when comparing sperm from 8-week males with sperm isolated from 22-week-old male mice.The authors interpret the observed changes as representative of the sperm DNA methylation changes that occur during normal chronological aging of the male. For an aged control group, the authors use sperm DNA of 22-week-old wild-type mates from the mTORC2 and mTORC2 KO breeding and compare the sperm methylation patterns found in sperm from those 22-week males to 8-week young males, that are intended to represent an old and a young cohort, respectively. DNA methylation analysis indicates that a disruption of mTORC2 (& decrease of BTB function) results in increased DNA methylation of sperm DNA, while a disruption of mTORC1 (and proposed increase of BTB tightness, not shown in the manuscript, though) resulted in increased hypomethylation.Weaknesses:While the hypothesis and experimental system are interesting and the data demonstrating the relevance of the mTORC2 complex for BTB function is convincing, several open questions limit the evidence that supports the hypothesis that the sperm DNA methylation changes seen in old males are caused by BTB failure following an imbalance of mTOR signaling complexes. The major critique points are the lack of a chronologically old group and the choice of 8 weeks & 22 weeks age of age:- Data illustrating the degree of BTB decline and sperm DNA methylation changes from chronologically "old" male mice is missing. 22-week-old mice are not considered old but are of good and mature breeding age, equivalent to humans in their mid-late twenties. In the manuscript, the 22-week-old wildtype mice show no evidence of BTB breakdown (Figure 3), so why are their sperm used to represent "aged" sperm?- Adding a group of "old" wild-type mice of 12-14 months of age, which is closer to the end of effective reproduction in mice, more equivalent to 45-59 year-old humans could be used to illustrate that (a) aging causes a marked decrease in BTB function at this time in mouse life, and that this BTB breakdown chronologically aligns with the age-associatedDNA hypermethylation seen in old sperm. Age-matched "old" mTORC1 KO, with a (supposedly) tighter BTB barrier, could then be expected to have a sperm DMA methylation profile closer to that of younger wild-type animals. Such data are currently missing. While the progressive testicular decline observed in the mTORC1 KO (Fig. 5) could make it difficult to obtain the appropriately aged mTORC1 KO tissues, it is completely feasible to obtain data from chronologically old wild-type males. (The progressive testicular decline further raises the question of what additional defects the KO causes, and how such additional defects would influence the sperm DNA methylation profile.) The addition of data from an old group to the currently included groups could strengthen the interpretation that the observations in the BTB-defective mTORC2 KO mice are modelling an age-related testicular decline, provided that the DMRs seen in the chronologically old group significantly overlap with the BTB-defective changes.- In the current form, the described differences in sperm DNA methylation are based on comparisons between pubertal mice (8 weeks) and mature but not old adult males (22 weeks), while a chronologically "old" group is missing from the data sets and comparisons. Thus, it appears that the described sperm methylation changes reflect developmental changes associated with normal maturation and not necessarily declining sperm quality due to aging. (Sperm obtained from 8-week-old mice likely were generated, at least in part, during the 1st wave of spermatogenesis, which is known to differ from the continuously proceeding spermatogenesis during the remained of the mature life. During the 1st wave of spermatogenesis, Sertoli cells are known to undergo gene expression changes which could contribute to varying degrees of BTB function, and thus have effects on the sperm DNA methylation profiles of such 1st wave sperm.)- It is unclear why the aging-related DMRs between the 8 and 22-week-old wild-type mice vary so dramatically between the two wild-type groups derived from the mTORC1 and the mTORC2 breeding (Fig. S4). If the main difference was due to mTORC1 or mTORC2 activity, both wildtype groups should behave very similarly. Changes seen in a truly "old" mouse (e.g. 20 weeks to 56 weeks), changes in "young mTORC1" and in "old mTORC2" are missing. How do those numbers and profiles compare to the shown samples?Some general comments regarding the chosen age of animals:- As mentioned, sperm from 8-week-old mice represent many sperm that were produced in the 1st wave of spermatogenesis; 22-week-old mice are not considered chronologically old mice, but mature and "relatively" young animals. 18-24 month-old mice are considered to be equivalent to 56-69 year-old humans, and might be more suitable to detect aging effects. "Old mice" for study purposes should be at least 12-14 months of age, ideally >18 months of age. 22 weeks (5 months of age) are mice at good breeding age, but still considered mature adults, not old males, and therefore are not expected to show typical aging health problems (like declining fertility).Even the cited reference (Flurkey et al. 2007) defines that "... mice used a reference group for "young mice" should be at least 3 months of age (~ 13 weeks), i.e. fully sexually mature. The authors specifically state: " The young adult group should be at least 3 months old because, although mice are sexually mature by 35 days, relatively rapid maturational growth continues for most biologic processes and structures until about 3 months. The upper age range for the young adult group is typically about 6 months. ... For the middleaged group, 10 months is typically the lower limit.... The upper age limit for the middleaged group is typically 14-15 months, because at this age, most biomarkers still have not changed to their full extent, and some have not yet started changing. For the old group, the lower age limit is 18 months because age-related change for almost all biomarkers of aging can be detected by then. The upper limit is 22-26 months, depending on the genotype." According to this reference, mice up to 6 months of age are generally considered "mature adults" (equivalent to humans 20-30 yrs), mice of 10-14 month are "middle-aged adults" (equivalent to ~38-47 human years) and 18-24 month mice are "old" (equivalent to human of 56-69 yrs.).Going on these commonly used age ranges, it is unclear why the authors used 8-week-old mice (generally considered pubertal to late adolescent age) as young mice and 5-month-old mice as "old mice".Differences seen between these cohorts most likely do not reflect aging, but more likely reflect changes associated with normal developmental maturation, since testis and epididymides continue to grow until about 10-11 weeks of age.- The DMRs identified between 8 and 22-week-old animals could represent DMRs that are dependent on developmental maturation more than being changed in an "age-dependent" manner (in the sense of increased chronological age). This interpretation is congruent with the fact that those DMRs are enriched for developmental categories.

- We are thankful to the reviewer for a detailed explanation of their disagreement with the ages of mice used in this study. In short, the reviewer suggests that our older group (22 weeks) is not old enough to represent aged animals and our young group (8 weeks) may still have spermatozoa from the first wave of spermatogenesis, and as such the observed differences between the 2 ages cannot be considered as aging-related but rather may represent different stages of maturation of the reproductive system. At the first glance this criticism looks valid.

However, to design our experiments we used our data that was not included to this manuscript initially. These data demonstrated that age dependent changes in sperm DNA are linearly or semi linearly associated with age in the age range from 56 to 334 days. Thus, within this interval any 2 ages, distant enough to register the difference in DNA methylation, can be used to assess age dependent changes in DNA methylation and changes in the rates of epigenetic aging of sperm in response to genetic manipulations. We have added these results now, - see “Identification of agedependent patterns in sperm DNA methylation” section in Material and Methods and “Patterns of age-dependent changes in sperm DNA methylation” in Results. We also consider that the reviewer’s suggestion that sperm from 8-week-old mice represents the first wave of spermatogenesis does not have ground. Indeed, C57BL/6 mice first have fertile sperm in cauda epididymis at 37 days of age [1], 19 days earlier than the age of 56 days (8 weeks) at which sperm was collected in our study in the youngest group of mice. Given that young C57BL/6 mice ejaculate spontaneously around 3 times per 5 days [2], 8 weeks old mice have ejaculated > 10 times since the first wave of spermatogenesis before the sperm was collected for our study, making negligibly small the chances of survival of any first wave sperm in their cauda epididymides to the age of 8 weeks. We have added this information to the text.

(1) Mochida, K.; Hasegawa, A.; Ogonuki, N.; Inoue, K.; Ogura, A. Early Production of Offspring by in Vitro Fertilization Using First-Wave Spermatozoa from Prepubertal Male Mice. J. Reprod. Dev. 2019, 65, 467–473, doi:10.1262/jrd.2019-042.

(2) Huber, M.H.; Bronson, F.H.; Desjardins, C. Sexual Activity of Aged Male Mice: Correlation with Level of Arousal, Physical Endurance, Pathological Status, and Ejaculatory Capacity. Biol. Reprod. 1980, 23, 305–316, doi:10.1095/biolreprod23.2.305.